# The WAVE Regulatory Complex Is Required to Balance Protrusion and Adhesion in Migration

**DOI:** 10.3390/cells9071635

**Published:** 2020-07-07

**Authors:** Jamie A. Whitelaw, Karthic Swaminathan, Frieda Kage, Laura M. Machesky

**Affiliations:** 1CRUK Beatson Institute, Glasgow G61 1BD, UK; K.Swaminathan@bradford.ac.uk (K.S.); L.Machesky@beatson.gla.ac.uk (L.M.M.); 2School of Chemistry and Bioscience, University of Bradford, Bradford BD7 1PD, UK; 3Department of Biochemistry and Cell Biology, Geisel School of Medicine at Dartmouth College, Hanover, NH 03755-3844, USA; Frieda.Kage@dartmouth.edu; 4Zoological Institute, Technische Universität Braunschweig, 38106 Braunschweig, Germany; 5Institute of Cancer Sciences, University of Glasgow, Glasgow G61 1QH, UK

**Keywords:** actin cytoskeleton, WAVE complex, stress fibers, focal adhesions, migration

## Abstract

Cells migrating over 2D substrates are required to polymerise actin at the leading edge to form lamellipodia protrusions and nascent adhesions to anchor the protrusion to the substrate. The major actin nucleator in lamellipodia formation is the Arp2/3 complex, which is activated by the WAVE regulatory complex (WRC). Using inducible *Nckap1* floxed mouse embryonic fibroblasts (MEFs), we confirm that the WRC is required for lamellipodia formation, and importantly, for generating the retrograde flow of actin from the leading cell edge. The loss of NCKAP1 also affects cell spreading and focal adhesion dynamics. In the absence of lamellipodium, cells can become elongated and move with a single thin pseudopod, which appears devoid of N-WASP. This phenotype was more prevalent on collagen than fibronectin, where we observed an increase in migratory speed. Thus, 2D cell migration on collagen is less dependent on branched actin.

## 1. Introduction

Cell motility is a highly regulated but dynamic process that is fundamental throughout biology. Many cells, including immune cells, pathogens, and cancerous cells, migrate and traverse biological barriers, often in response to a stimulus. As such, a hallmark of cancer metastasis is the ability of the cell to invade surrounding tissues following migration from the primary tumour site [1]. Classical 2D mesenchymal migration is defined by the formation of a protrusive front and retracting rear. This is predominantly driven by the Arp2/3 complex that induces the nucleation of branched actin filaments at the membrane to form a broad, flat protrusion at the leading edge, termed a lamellipodium [2]. These newly forming lamellipodia are constantly interacting with the surrounding matrix and relaying signals through integrin-based adhesions, a process commonly known as mechanotransduction [3]. Signalling mechanisms, including those linked to myosin-based contractility, result in the release of focal adhesions at the rear of the cell and the recycling of adhesion receptors back to the front of the cell.

In most circumstances, cell migration relies on the well-ordered assembly and disassembly of actin filaments to generate localised plasma membrane protrusions such as lamellipodia, filopodia, and blebs. For example, during lamellipodia formation, the activation of the small GTPase Rac1 stimulates the Arp2/3 complex through the Scar/WAVE protein [4], which is embedded in a complex known as the WAVE regulatory complex (WRC), consisting of NCKAP1 (HEM1), CYFIP1/2 (Sra1/PIR121), Abi1-3, HSPC300 (Brick1) and WAVE1-3 (Scar1-3) [5]. In the WRC, active Rac1 interacts with Cyfip1 at two different positions [6,7], which is proposed to lead to a conformational change in the WRC and release of the auto-inhibited VCA domain of WAVE, which subsequently actives the Arp2/3 complex. This cascade also results in feedback-loops, which can locally restrict protrusions through proteins such as CYRI-B and Arpin with the WRC or Arp2/3 complex, respectively [8,9].

As lamellipodia form, integrins engage extracellular matrix (ECM) ligands, which in turn trigger signaling events through focal adhesion kinase and paxillin to recruit GEFs, such as GIT1 and β-PIX for Rac1. This activation induces actin polymerisation through the Arp2/3 complex and stimulates further elongation of focal adhesions [10].

While the role of the WRC is well understood in the regulation of branched actin dynamics in lamellipodia, how crosstalk occurs with focal adhesion formation and dynamics is less well understood. Previous work from our lab indicated that disruption of the WRC alters signaling through focal adhesion kinase and promotes cancer cell invasion [11]. Silva et al. found that WRC knockdown cells displayed larger focal adhesions connected to actin stress fibers [12]. Furthermore, the WRC has been implicated in internalisation and recycling of α5β1 integrins from the perinuclear space [13].

To help understand more about the role that the WRC has on adhesion and cell migration, we deleted *Nckap1* in mouse embryonic fibroblast (MEF) cells. There is a significant body of evidence describing the functions of the WRC complex. However, these systems do not faithfully reflect the normal physiological functions of the WRC complex. In vivo functional studies using the mouse model are hampered by prenatal lethality phenotypes of WRC complex members. Therefore, we established an inducible *Nckap1* floxed mouse model to robustly isolate MEF for WRC functional studies. NCKAP1 belongs to the HEM family of proteins, originally thought to be transmembrane proteins, but now known to be cytoplasmic and is conserved as a subunit of the WRC in a wide range of organisms. It has been implicated in a wide range of cytoskeletal functions, including embryonic development [14], axonal growth [15], differentiation of neurons [16], and chemotaxis [17]. Here we show that cells lacking NCKAP1 change from lamellipodia-based to pseudopodia-like migration that has altered focal adhesion dynamics and reduced migration speed/distance that can be partially rescued by plating on collagen.

## 2. Materials and Methods

### 2.1. Transgenic Mice and Isolation of Nckap1^fl/fl^ Mouse Embryonic Fibroblasts

All animal experiments were performed according to the UK Home Office regulations and in compliance with EU Directive 2010/63 and the UK Animals (Scientific Procedures) Act 1986. All protocols and experiments were previously approved by the Animal Welfare and Ethical Review Body (AWERB) of the University of Glasgow and were accompanied by a UK Home Office project license (7008123—July 2014; PE494BE48—April 2019). The *Nckap1* floxed mouse strain was created using a targeting vector (PG00182_Z_4_C05) obtained from the consortium for The European Conditional Mouse Mutagenesis Program (EUCOMM) and described [18]. ES cells transfections, clone selection, and injection into C57BL/6J blastocysts were performed according to standard protocols outlined in [19,20]. *Nckap1^fl/fl^* mice were bred with *Rosa26:CreER^t2+^* [21] and *Ink4a*^−/−^ [22] to obtain the genotype *Rosa26:CreER^t2+^*; *Ink4a*^−/−^; *Nckap1^fl/fl^*. Heterozygous matings were set up for embryos. Genotyping was performed by Transnetyx (Cordova, TN, USA). Mouse embryonic fibroblasts were harvested from a pregnant female mouse at day E13.5 following the protocol outlined in [23].

### 2.2. Generation of Nckap1 KO B16-F1 Cells

*Nckap1* knockout in B16-F1 mouse melanoma cells was essentially carried out as described in [24,25]. In contrast to *Nckap1* KO clones (#6 and #21) used in Dolati et al., which still formed low numbers of aberrant lamellipodia due to compensatory expression of the hematopoietic counterpart *Nckap1L*, *Nckap1* KO clone #16 that was virtually devoid of lamellipodia was used in this study.

### 2.3. Mammalian Cell Culture Conditions

Mouse embryonic fibroblasts and mouse melanoma B16-F1 cells were maintained in Dulbecco’s Modified Eagles Medium (DMEM) supplemented with 10% FBS, 2 mM l-glutamine. Mouse embryonic fibroblasts were maintained in complete DMEM supplemented with 1 mgmL^−1^ primocin.

### 2.4. Transfection of Mammalian Cells Lines

*Nckap1^fl/fl^* mouse embryonic fibroblasts were transiently transfected by electroporation (Amaxa, Kit T, program T-020) with 5 μg DNA and plated overnight to recover.

B16-F1 cells were plated on a 6-well plate and grown to 70% confluency and later transfected with Lipofectamine 2000 following the manufacturer’s guidelines with 2–5 μg DNA.

### 2.5. Genetic Knockouts

Inducible knockout *Nckap1^fl/fl^* MEFs were generated by the addition of 1 μM 4-hydroxytamoxifen (OHT) in the growth medium being replaced every 3 days over a 7-day period.

### 2.6. Analytical PCR

gDNA was isolated from DMSO or OHT treated *Nckap1^fl/fl^* MEFs using a Qiagen DNeasy Blood and Tissue kit following the manufacturer’s protocol. PCR was performed to determine the efficiency of recombination by the loss of the N-terminal region of *Nckap1* using specifically designed primers (#fw: CTCTCTTGTCTACTGTGCAGG and #rv: CTCGTAGACCAAACTAGCCTCAAG).

### 2.7. Cell Proliferation and Viability

Cells were harvested and adjusted to 1 × 10^4^ cells, which were plated onto 6-well plates. Each subsequent day the cells were harvested and counted using a hemocyotometer for cells per well to determine the proliferation rates of the cell lines. Data are presented from 3 technical replicates, repeated three times independently.

On days 3, 5, and 7 after plating, harvested cells were also tested for their viability using Trypan Blue solution. A 1:1 cell suspension of cells and 0.4% Trypan Blue was mixed and added to the hemocytometer and left for 2 min prior to counting. Viable cells do not take up the dye, while dead cells are permeable to the dye. Counts were adjusted as a percentage of live/dead from 3 technical replicates, repeated three times independently.

### 2.8. SDS-PAGE and Western Blotting

Lysates were collected on ice by scraping cells in RIPA buffer (150 mM NaCl, 10 mM Tris-HCl pH 7.5, 1 mM EDTA, 1% Triton X-100, 0.1% SDS, 1X protease and phosphatase inhibitors). The tubes were centrifuged for 10 min at 15,000 rpm and 4 °C. The lysate was transferred to a clean Eppendorf tube and measured using Precision Red.

The 40 μg of protein lysate was resolved on NuPAGE Novex 4–12% Bis-Tris gels and transferred onto a nitrocellulose membrane Bio-Rad system. Membranes were blocked with 5% BSA in TBS-T (10 mM Tris pH 8.0, 150 mM NaCl, 0.5% Tween-20) for 20 min prior to overnight incubation with the primary antibody at 4 °C on a shaking incubator. Membranes were then washed three times for 5 min each in TBS-T. Membranes were then incubated at room temperature for 1 hour with secondary AlexaFluor conjugated antibodies, after which the blots were washed again for 5 min in TBS-T three times before being imaged on the Li-Cor Odyssey CLx machine. Images were then analysed using the Image Studio Lite Version 5.2. Lysates were harvested on 4 independent occasions from *Nckap1^fl/fl^* and *Nckap1*^-/-^ MEFs.

Antibodies used: NCKAP1 (Abcam, Cambridge, UK: ab126061), CYFIP1 (Abcam: ab156016), WAVE2 (Santa Cruz Biotechnology, Dallas, TX, USA: sc-33548) and α-Tubulin (Sigma, St. Louis, MO, USA; clone B512) as the loading control.

### 2.9. Immunofluorescence Analysis

Cells were plated onto sterile 13 mm glass coverslips that had been previously coated with 10 μgmL^−1^ Rat tail Collagen I, 10 μgmL^−1^ fibronectin, 10 μgmL^−1^ laminin, 10 μgmL^−1^ Poly-l-Lysine or 50% FBS diluted in sterile PBS. Cells were fixed with 4% paraformaldehyde for 10 min, RT. Coverslips were then washed three times with PBS before incubation with blocking buffer (0.05% Triton X-100, 5% BSA, PBS) for 15 min, rotating. Primary and secondary antibodies were diluted in blocking buffer and incubated with the coverslips in a dark, humidified chamber for 1 h. Coverslips were washed three times in PBS and once in MilliQ water before mounting with FluoromountG solution containing DAPI.

Imaging was conducted on either a Zeiss LSM 880 Airyscan confocal microscope with a heated incubator or a Nikon A1R confocal microscope with a heated stage incubator.

Antibodies used: WAVE2 (Santa Cruz: sc-33548), p34 ArpC2 (Millipore, Burlington, MA, USA; 07-227-1), Cortactin (Millipore; clone 4F11), α-Tubulin (Sigma; clone B512), phalloidin (ThermoFischer Scientific, Waltham, MA, USA), N-WASP (Sigma; HPA005750), Vinculin (Sigma; clone hVIN-1), Paxillin (BD Biosciences clone 349, Franklin Lakes, NJ, USA) and DAPI (Southern Biotech, Birmingham, AL, USA).

### 2.10. Lamellipodia Formation

An immunofluorescence analysis for lamellipodia formation was conducted as described above. Images were scored for the presence or absence of lamellipodia and presented as a percentage. For the *Nckap1^−/−^* cells, this was further categorised into cells that had formed a protrusion or remained round in shape, also presented as a percentage.

### 2.11. xCELLigence

E-plate 16 were coated with collagen I overnight and equilibrated with DMEM complete for 30 min prior to image at 37 °C. Cells were harvested and adjusted to 5 × 10^3^ per well. The cells were seeded in technical quadruplicate, and the plate was immediately transferred to the Acea RTCA DP xCELLigence machine maintained at 30 °C, 5% CO_2_. The cell index was measured at 5-min time intervals for 8 h and readings were averaged for each condition. The impedance between the electrodes and cells determined cell index over time. Data are presented from 3 independent replicates.

### 2.12. Focal Adhesions

Mouse embryonic fibroblasts were plated onto collagen-coated coverslips and left to adhere and migrate overnight. These were fixed and stained for AlexaFluor Phalloidin (ThermoFischer Scientific) and Mouse anti-Paxillin (BD Biosciences, clone 349) or Mouse anti-Vinculin (Sigma; clone hVIN-1) to measure cells area and focal adhesions, respectively. Z-stacked images were taken on the Zeiss 880 confocal microscope and analysed using Fiji software Version 2.0.0, where the focal adhesions were enhanced using a Gaussian blur and identified using ImageJ’s find maxima within tolerance. These were then measured using the analyze particles. The output image from the ImageJ-derived maxima is overlaid onto a greyscale image of the FAs from the original file to indicate that the method is able to distinguish the majority of focal adhesion proteins from the original image. Data are presented from two independent experiments.

### 2.13. Focal Adhesions Turnover

MEFs were transiently transfected with LifeAct-TagRed and pEGFP-Paxillin as described above and plated onto 35 mm glass-bottom Ibidi dishes coated with collagen I or fibronectin. Short movies of 1 frame per minute for 30 min were obtained using the 488 nm and 568 nm laser on the Zeiss LSM 880 confocal microscope with Airyscan module at 37 °C and 5% CO_2_. Time-lapse movies were processed using Fiji software to stabilize and submitted to the Focal adhesion analysis server (FAAS) (http://faas.bme.unc.edu/) [26]. A threshold of 2.5 units was maintained across all image sets, and positive structures or 15 pixels^2^ that last for at least 5 consecutive frames were quantified as being a focal adhesion. The server is set up to calculate assembly and disassembly rates using the same method as described in Reference [27] with the modifications described in Reference [28]. Briefly, the software follows the mean intensity of Paxillin through time and fits a linear model to the log-transformation time series of the Paxillin intensity as described in Reference [28]. FAAS also calculates dynamic properties such as focal adhesion longevity (lifetime) from the birth and death of focal adhesions between frames described in Reference [28]. Data are presented from 3 independent experiments.

### 2.14. Actin Photoactivation—Retrograde Flow

Photoactivation of actin and retrograde flow analysis was conducted as described in Reference [29]. Briefly, B16-F1 cells were transiently transfected with LifeAct-TagRed and PA-GFP-Actin as described above. Imaging was conducted on a Zeiss 880 confocal microscope using a Plan-Apochromat 63×/1.4 oil DIC objective lens. The PA-GFP-Actin and LifeAct-TagRed were monitored with 488 nm and 568 nm lasers, respectively. A single pulse with a 405 nm laser (pulse length *t* = 0.5 s) obtained photoactivation of actin at the ROI. Acquisitions were taken every second for 60 frames with an initial 5 s to obtain baseline GFP intensity prior to activation. Data are presented from 3 independent experiments.

### 2.15. Stress Fiber Quantification

MEFs were plated onto glass bottom dishes coated with Collagen I and incubated overnight at 37 °C and 5% CO_2_. The coverslips were fixed and stained with AlexaFluor Phalloidin as described above. Z-stacked images obtained from the Nikon A1R microscope. Images were processed using the macro to max project the z-stack, highlight the stress fibers with a Difference of Gaussians threshold and Ridge Detection to identify and quantify stress fibers. In the control cells, we manually remove the lamellipodia actin signal that is intensified by the Difference of Gaussians thresholding. Data are presented from 3 independent experiments.
run(“Z Project...”,“projection = [Max Intensity]”);run(“Difference of Gaussians”, “ sigma1 = 4 sigma2 = 0.50 enhance”);setOption(“ScaleConversions”, true);run(“8-bit”);setAutoThreshold(“Default dark”);setThreshold(40, 255);setThreshold(40, 255);//setThreshold(40, 255);setOption(“BlackBackground”, true);run(“Convert to Mask”);run(“Ridge Detection”, “line_width = 3.5 high_contrast = 230 low_contrast = 87 estimate_width extend_line displayresults add_to_manager method_for_overlap_resolution = NONE sigma = 1.51 lower_threshold = 3.06 upper_threshold = 7.99 minimum_line_length = 15 maximum = 0”);

### 2.16. Rac1 Photoactivation

Photoactivation of Rac1 was as described in Reference [9]. Briefly, *Nckap1^fl/fl^* MEFs were transiently transfected by electroporation (Amaxa, Kit T, program T-020) with 5 μg of photoactivatable Rac1 plasmid [30] (pTriEX-LOV2-Ja-Rac1-mCherry) purchased from Addgene, plasmid #22027. Cells were maintained in the dark and starved with serum-free medium overnight prior to imaging. Imaging was conducted on a Zeiss 880 confocal microscope maintained at 37 °C, 5% CO_2_. Time-lapse imaging was conducted for 150 frames with 2 s intervals. An initial 1-min of imaging (29 Frames) was collected with the 568 nm excitation to monitor baseline Rac1 activity. Photoactivation of a 100-pixel diameter ROI equating to 136.4 µm^2^ circle using pulses of 458 nm laser started at frame 30 and continued for each frame to 150. Movies were processed as described in Reference [9]. Endpoint overlay images are derived from the differences between time-points *t* = 30 (start of photoactivation) and *t* = 180 (end). The whole cell is shown in grey and protrusions in cyan and the retraction area in magenta. Data presented from >3 independent experiments.

### 2.17. Random Migration Assay

Cells (1 × 10^5^) were plated onto a 6-well glass bottom plates coated with ECM components as described above. Wide-field images were captured every 10 min for 16 h using a Nikon Eclipse TE2000 microscope with a Plan Fluor 10 X objective equipped with a heated CO_2_ perfused chamber. Images from 2 independent experiments were analysed using Fiji software where individual cells were tracked using the mTrackJ plugin. Spider plots were generated using the chemotaxis and migration tool plugin.

### 2.18. Pseudopodia Quantifications

From the random migration assay, we calculated pseudopodia extension parameters in *Nckap1*^−/−^ KO MEFs migrating over collagen or fibronectin. Percentages of pseudopodia-like extensions were determined from the number of cells that form pseudopodia-like extensions over the 16-hour experiment. Pseudopodia length was determined by measuring the length of extensions in cells every hour from three random migration 16-h movies on each substrate. Pseudopodia length was taken from the edge of the nucleus to the pseudopodia tip as a direct line using ImageJ. Pseudopodia-extension lifetimes were quantified from all cells that form at least 1 pseudopodia over the 16-h experiment. In some instances, more than one pseudopodium was produced by the cell, but this was still calculated as a single event until the cell stopped migrating and the pseudopodia were retracted. These were correlated to speed or distance travelled while in a pseudopodia shape.

### 2.19. Statistics and Reproducibility

All datasets were analysed using GraphPad Prism version 8. Datasets were tested for normality and then analysed using the appropriate statistical test, as described in each figure legend. Significance levels rejecting the null hypothesis are represented above figures where: NS *p* > 0.05, * *p* < 0.05 *, ** *p* < 0.01, *** *p* < 0.001 and **** *p* < 0.0001. Where significance was not reached, nothing was added above the graphs.

## 3. Results

### 3.1. Generation of Inducible Nckap1 Knockout Mouse Embryonic Fibroblasts

We isolated mouse embryonic fibroblasts with the genetic background expressing *Rosa26:CreER^t2+^*; *Ink4a^−/−^*; *Nckap1^fl/fl^*. Using the Cre-ER^t2^ system, we were able to genetically delete *Nckap1* upon Cre mediated recombination through the addition of 4-hydroxytamoxifen (OHT). Deletion occurs at the N-terminal region of *Nckap1*, resulting in a truncated non-functional gene (Appendix A), which was analysed by analytical PCR 5 days post OHT treatment (Appendix A). Disruption of *Nckap1* results in the loss of NCKAP1 protein after 7 days of OHT treatment along with the other members of the WRC; CYFIP1, WAVE2 (Figure 1B), confirming the requirement of NCKAP1 for the integrity of the WRC [11,25,31]. Loss of NCKAP1 severely retarded cell proliferation as early as day 2, post OHT treatment (Appendix A). However, these cells were still viable using a live/dead marker stain at 7 days post OHT treatment, when the levels of NCKAP1 are undetectable by immunoblot (Figure 1B).

### 3.2. NCKAP1 Is Required for Fibroblast Spreading

The WRC has been long known to be fundamental for lamellipodia formation, first demonstrated for the Scar/WAVE protein alone [4] and later for several subunits of the WRC [31]. We confirmed that in fibroblasts, WAVE2, ArpC2 (a member of the Arp2/3 complex), and Cortactin (a regulator of the Arp2/3 complex) (Reviewed in [32]) localised to the leading edge of the lamellipodia in control cells. As expected, the deletion of *Nckap1* resulted in a loss from the cell periphery and a diffuse staining throughout the cytoplasm (Figure 1A). We did not observe many filopodia-like structures in control MEFs, whereas *Nckap1* KO MEFs (*Nckap1^−/−^*) had numerous filopodia-like structures containing cortactin (Figure 1A), similar to what has been shown for filopodia in growth cones [33].

While analysing the localisation of proteins involved in lamellipodia formation, we observed various shapes of *Nckap1* KO cells. Scoring cell shapes revealed that almost 90% of control *Nckap1* MEFs (*Nckap1^fl/fl^*) presented with lamellipodia (Figure 1C,D). *Nckap1* KO MEFs were completely lacking any lamellipodial protrusions and had mostly small and rounded with numerous filopodia extensions. Around 30% of *Nckap1* KO cells displayed an elongated phenotype, with structures resembling collapsed protrusions enriched with microtubules (Figure 1C,D). We next examined the ability of *Nckap1* KO cells during spreading to determine the cell size changes over time. Control and *Nckap1* KO MEFs were added to coverslips coated in laminin, fibronectin or collagen, which engage a variety of integrins and promote cell migration [34]. However, *Nckap1* KO MEFs do not spread on these surfaces and remain significantly smaller in area when compared to the controls MEFs (Figure 1E). We also tested cell area on poly-l-lysine (PLL) and fetal bovine serum (FBS) where integrin activation is not a requirement for attachment [35]. Under these conditions, the control cells were still able to spread, presenting with similar cell areas as the observed on the other ECM components (Figure 1E). However, *Nckap1* KO cells remained predominantly rounded with a significantly smaller cell area than control cells in all conditions tested (Figure 1E). *Nckap1* KO spreading on collagen and fibronectin appeared greater than other ECM components, with a larger range of cells spreading on collagen (Figure 1E). We also used xCELLigence (ACEA Biosciences Inc., San Diego, CA, USA) to measure cell spreading continually in real-time [36]. Control MEFs spread rather slowly for around 40 min in the adhesion phase and then showed a sharp increase in cell spreading for around 2 h after which they plateau [36] (Figure 1F). However, *Nckap1* KO MEFs do not spread significantly and remain at around 20% the size of the control cells (Figure 1F), similar to the overnight spreading assay in Figure 1E.

The small GTPase Rac1 is required for lamellipodia formation through its interactions with the WRC [37]. A previous study showed that microinjection of active Rac1 in *Nckap1* knockdown cells failed to induce lamellipodia [31]. We further queried whether activation of Rac1 could trigger any changes in cell area or extension of the plasma membrane using MEFs expressing a photoactivatable Rac1 (PA-Rac1) probe [30]. We used a series of 458 nm laser pulses at the edge of the cell to induce the formation of a lamellipodium. In the case of the *Nckap1* KO cells that do not have a distinct leading edge, we illuminated a region at an edge of round cells or various points on the protrusion of pseudopod-like extensions. Activation of Rac1 in control MEFs induced lamellipodia formation and membrane ruffles as similarly described in References [9,30] (Figure 1G–I). There was also substantial lateral spread from the activation site in control MEFs (Figure 1H,I). Contrastingly, the *Nckap1* KO MEFs did not induce any detectable lamellipodia or membrane ruffles. Thus, a functional WRC is a requirement for Rac1 induced lamellipodia establishment, in agreement with previous studies using microinjection of active Rac1 [31].

### 3.3. Loss of NCKAP1 Alters the Actin Dynamics in the Cell

Within the lamellipodium, polymerised actin filaments mostly orient with their barbed/plus ends facing the cell edge [38]. The resulting polymerisation of actin at the edge generates a force that pushes the membrane forwards while the actin filaments drive inward, creating a retrograde flow of actin from the lamellipodium through to the lamella region of the cell. This flow is inhibited by the formation of focal adhesions [39] in the lamellipodia region, where the actin filaments are bundled to form stress fibers [40]. Focal adhesion engagement acts as a molecular clutch to translate the backward flow into pushing force to drive the cell membrane forward.

To test whether a retrograde flow will still occur in the absence of the WRC, we measured the actin dynamics in *Nckap1* KO cells using photoactivatable-GFP-actin (PA-GFP-Actin) in migrating B16-F1 cells. These were kindly gifted by Professor Klemens Rottner (Described in Reference [24,25]) and were used here due to their ability to form consistent stable, broad lamellipodia for the duration of the experiment. A retrograde flow of PA-GFP-Actin was observed in B16-F1 control cells from the leading edge to the lamella within the range of 13.06 ± 0.65 seconds (Figure 2A–C; Appendix A), similar to the observations of actin flow in the 1st half of a lamellipodia region [41]. In contrast, the PA-GFP-Actin once activated at the leading edge of *Nckap1* KO cells decayed much slower (41.02 ± 15.71 s) and within the time frame of the experiment did not traverse rearwards (Figure 2A–C; Appendix A). While the majority of PA-GFP-Actin remained at the activation site (Figure 2C; Appendix A), there were some instances where a flow laterally along actin cable/stress fibers was observed, indicating that the polymerised actin is likely shifted into linear bundled actin filaments. This decay of PA-GFP-Actin in the *Nckap1* KO cells is reminiscent of cells treated with Jasplakinolide, an actin stabilisation drug [29].

Comparable to WAVE knockdown cells [12], we observed thick actin fibers around the periphery of the *Nckap1* KO MEFs. These appear to connect to peripheral focal adhesions. As there was little to no retrograde flow in the *Nckap1* KO cells but many thick stress fibers, we set out to quantify these stress fiber parameters, such as length and thickness. Here we analysed z-projections of phalloidin-stained cells acquired from a confocal microscope. Indeed, only a few peripheral stress fibers were detected within *Nckap1* KO cells (Figure 2F). These were shorter (Figure 2D) but significantly thicker, encompassing the *Nckap1* KO cells (Figure 2E), similar to those observed in WAVE knockdown cells [12].

With altered actin dynamics, we next asked whether another actin nucleating factor compensates for the loss of the WRC in promoting Arp2/3 activation. Previous work suggested that N-Wasp could rescue WRC depletion and promote 3D migration and invasion [11]. Using a specific N-WASP antibody, we analysed the endogenous localisation of N-WASP in *Nckap1* KO cells that have a pseudopod-like protrusion (Figure 2G). In *Nckap1* KO MEFs, we observed no N-WASP localisation at the pseudopod-like extensions, indicating that in MEFs under the conditions that we tested here, N-WASP cannot compensate for the loss of the WRC, in contrast to what occurs in *Dictyostelium* [42].

### 3.4. Loss of the Scar/WAVE Complex Affects Focal Adhesion Dynamics

Cell migration involves the rearrangement of the actin cytoskeleton and adhesions linking the extracellular matrix to the actin cytoskeleton. Nascent focal complexes form within the lamellipodia of migrating cells [43], eventually maturing to focal adhesions (FAs), reviewed in Reference [44]. The assembly and maturation process of FAs is a highly ordered process where focal complexes form through integrin engagement with the ECM at the leading edge of the lamellipodium. At this stage, the complex is immature and forms weak connections to actin filaments through adaptor proteins such as talins. Many of these are unstable and readily disappear. However, the recruitment of additional scaffolding proteins and phosphorylation by focal adhesion kinase (FAK) promotes maturation, and more stable FAs will begin to elongate along the direction of the retrograde flow, causing traction to the substrate ECM, slowing down the actin retrograde flow, and inducing stress fiber formation [39].

Recently, it was shown that the WRC is responsible for cell adhesion to the matrix for both macrophages and B16-F1 melanoma cells, where mutants for the WRC are slower to adhere and stay adhered for longer to the ECM [25]. We noticed a similar strong adherence of *Nckap1* KO MEFs during normal tissue culturing conditions where the cells require longer trypsin treatment to detach (data not shown). Indeed, the involvement of the WRC and Arp2/3 complex in focal adhesion formation has been reported in various systems. The Arp2/3 complex was shown to interact transiently with focal adhesion proteins such as vinculin [45] and with FAK via FERM-mediated interactions [46]. The depletion of the WRC has been reported to have an impact on focal adhesions in both 2D and 3D migration such as increased paxillin phosphorylation through FAK signalling [11,25]; adhesion sizes are altered [12] and cell-cell junctions are impaired [47]. We found that *Nckap1* KO MEFs have fewer adhesions (Figure 3A,C) similar to observations of WAVE knockdown cells [12]. *Nckap1* KO MEFs displayed significantly smaller adhesions than control MEFs (Figure 3B), unlike some other studies reporting larger focal adhesions [11,12]. Paxillin staining also confirmed this result (Figure 3D) and these focal adhesions align well with the tips of thick actin stress fibers [12,48]. We speculate that the accumulation of focal adhesion proteins at the tips of the protrusions may help in anchoring the front of the protrusion during migration.

We also examined the dynamics of these FAs, measuring formation and disassembly rates based on live imaging with pEGFP-Paxillin and LifeAct-TagRed over a 30-min time course (Figure 3E; Appendix A). While analysing the adhesion turnover, we observed that there was a distinct lack of centrally located FAs in the *Nckap1* KO cells. We also observed a strong cage-like actin network around the nucleus of these cells (Figure 3I). Using an online tool, the focal adhesion analysis server (FAAS), to quantify adhesion dynamics, we found that overall focal adhesion turnover is slower in *Nckap1* KO MEFs compared to control cells (Figure 3F–H). While the assembly rates between the control and *Nckap1* KO cells were similar on the respective surface, the disassembly rates of focal adhesion proteins were much slower in the *Nckap1* KO MEFs (Figure 3G), resulting in a significant increase in the lifetime of focal adhesions for *Nckap1* KO cells on fibronectin (Figure 3H). Interestingly, focal adhesion turnover for both control and *Nckap1* KO MEFs was higher on collagen I coated dishes than fibronectin (Figure 3E,F).

### 3.5. Nckap1 KO Cells Can Migrate without Lamellipodia on Collagen

There is a direct link between actin turnover, adhesions, and cell migration over planar substrates. Cells with high migratory speeds such as *Dictyostelium discoideum*, neutrophils, and fish keratocytes have a fast actin turnover and weak adhesions. Contrastingly, many mammalian cells from tissues have a characteristically low migratory speed when compared to the above cells. This is primarily due to strong adhesion and slower actin turnover. Using random migration assays, we found that control cells migrated at average speeds of 0.44 ± 0.12 μm/min over fibronectin and 0.53 ± 0.18 μm/min over collagen (Figure 4A), whereas there is around a 3-fold reduction in the average speed of *Nckap1* KO cells on fibronectin, but less than a 2-fold reduction on collagen (0.149 μm/min on FN and 0.295 μm/min on collagen) (Figure 4A). *Nckap1* KO cells also migrated significantly further on collagen than fibronectin (Figure 4B,C; Appendix A). We next analysed whether *Nckap1* KO cells migrating over collagen transitioned between the two phenotypic shapes described in Figure 1C and if this affected cell migration speeds. We observed that many *Nckap1* KO cells remained round and barely moved throughout the experiment. While in the rounded shape, cells remain relatively immobile, which correlated with reduced speeds and migratory distance (Figure 4D,E; Appendix A). A few cells were able to transition from a rounded shape to form a pseudopod-like protrusion and were able to increase their speeds and migrate further during this time (Figure 4D,E; Appendix A). This ability of the *Nckap1* KO cells to polarize and form a protrusion and thus migrate is observed on both fibronectin and collagen-coated surfaces; however, this migratory mode was more prevalent on collagen-coated surfaces where a higher percentage of cells formed pseudopod-like extensions (Figure 4F). Many *Nckap1* KO cells form and extend more than one pseudopod. In some instances, the pseudopod may split, but it was observed that for effective migration, one pseudopod would dominate. These pseudopodia extensions formed while migrating over collagen were also significantly longer in length than those formed on fibronectin (Figure 4G). Interestingly, the pseudopod-like extensions produced by the *Nckap1* KO cells had a similar range of lifetimes on both surfaces (Figure 4H,I). However, while in a pseudopod shape, *Nckap1* KO cells had improved migratory capabilities over collagen, where the pseudopod-like extensions correlated with increased speed (Figure 4H) and migratory distance (Figure 4I), when compared to fibronectin. From DIC imaging (Figure 4E; Appendix A), these cells appear to polarise and produce a long, thin protrusion that strongly adheres at the tip of the protrusion, correlating with the accumulation of focal adhesion proteins we observe at the tips of these cells (Figure 3E). Similar to lamellipodial-based migration, *Nckap1* KO cells adhere at the front and the rear is pulled towards the leading edge. This is highly akin to studies of cell migration in a 3D collagen [49]. Overall our data indicate that the rounded cells represent a severe phenotype with a majority of these cells remaining relatively inactive. We speculate that the actin stress fibers and slower focal adhesion turnover results in a strong actin cytoskeleton that has to be overcome to transition from the rounded to a pseudopod-like protrusive phenotype for migration.

## 4. Discussion

In this study, we further explored the relationship between NCKAP1 and adhesion. Our study confirmed previous findings that NCKAP1, (one of the two largest subunits of the WRC) is essential for the WRC’s function in driving Arp2/3 mediated actin polymerisation at the leading edge [11,31]. However, other family members similar to WAVE, such as the ubiquitous neural-WASP (N-WASP) can compensate for the loss of the WRC and promote 3D migration and invasion of cancer cells [11] and also can substitute for WRC to rescue *Dictyostelium discoideum* migration [42,50]. When both Scar and WASP are deleted in *Dictyostelium*, they become immobile but the cells generate numerous filopodia due to excess activity of the formin dDia2 [51]. We did not test for mDia2, the mammalian equivalent of dDia2, but we were unable to observe N-WASP localisation in the tips of the pseudopod-like protrusions, indicating that at least for MEFs and B16-F1 cells migrating over 2D substrates, N-WASP does not detectably compensate for the loss of the WRC in activating the Arp2/3 complex at the leading edge to promote lamellipodia or pseudopod formation.

As it was recently shown, in the absence of the Scar/WAVE complex, WASP requires active Rac1 to drive pseudopod extensions in *Dictyostelium* [42]. However, the mammalian counterpart to *Dictyostelium* WASP, N-WASP is activated by both Rac1 and Cdc42 [52], and *Dictyostelium* does not express Cdc42. This one difference between *Dictyostelium* and mammalian cells may be why there is no compensation by N-WASP in 2D migration. Indeed there is plentiful active Rac1 in WRC depleted cells, especially after Rac1 activation or microinjection of active Rac1 that would have the ability to drive pseudopodia formation as seen in [42]. Therefore, something else is preventing this process in mammalian cells, possibly the high adhesiveness of mammalian cells compared to *Dictyostelium* [53] keeping the cells anchored to the substrate. Perhaps the WRC provides some specific link between adhesion and lamellipodia protrusion that isn’t compensated in mammalian cells by N-WASP.

MEFs lacking a WRC appear to transition their actin into thick cables surrounding the periphery of the cell. These thick cables of actin in rounded cells do not lead to protrusion of the cell edge, even after photoactivation of Rac1 (Figure 2). Whereas, photoactivation of Rac1 caused extensive lamellipodia formation in CYRI-B mutant cells, which were unable to regulate the WRC [9]. Recently it was shown in *Dictyostelium discoideum* that active Rac1 was able to drive pseudopod extensions even in the absence of the WRC [42]. However, in MEFs we were unable to induce pseudopod extensions in the absence of a functional WRC through Rac1 activation alone. This result was similar to Steffen et al., who microinjected active Rac1 into cells with depleted WRC [31]. It remains unclear why in some cells (e.g., *Dictyostelium*) N-WASP can compensate for the loss of WRC, but not in other cells (e.g., fibroblasts, B-16 melanoma cells- this study). It could be interesting to explore this further in rapidly moving mammalian cells, such as macrophages.

Actin dynamics have long been implicated in focal adhesion turnover [54]. While there are studies directly linking the Arp2/3 complex to focal adhesions [45,46], less is known about the role of the WRC. Previously it was shown that WRC-depleted cells had an increase in FAK signalling and a reduction in focal adhesion dynamics [11]. While there are parallels between *Nckap1* KO cells migrating in 2D and cells migrating in 3D collagen matrices is nice, there are also some differences in structures such as focal adhesions. Notably, focal adhesions tend to be smaller and distributed over the surface of the cell in contact with the matrix, rather than the large elongated or arrow-head shaped structures commonly observed in 2D on rigid substrates. Additionally, focal adhesion proteins in 3D regulate cell speed through matrix deformation [55]. In 2D on glass, FAK signalling promotes phosphorylation of paxillin that enhances lamellipodia protrusions and cell migration [56]. Here we show that paxillin-containing focal adhesions are significantly smaller in MEFs depleted for *Nckap1* and their turnover is impaired (Figure 3). Indeed, there is evidence that other actin nucleating proteins such as formins (mDia1 and mDia2) are associated with actin dynamics and focal adhesion turnover. mDia1 induces the formation of stress fibers and focal adhesions through associating with ROCK [57,58]. Similar to disruption of *Nckap1*, the inhibition of the formin (mDia2) depresses lamellipodia dynamics, causes smaller focal adhesions that have reduced disassembly rates coalescing to have impaired cell velocity [59]. Thus, actin turnover in lamellipodia is intimately linked to focal adhesion dynamics, likely by various mechanisms that may involve retrograde flow and both branched actin and formin-mediated unbranched actin.

We hypothesise that the increased ability of the *Nckap1* KO cells to form the pseudopod-like protrusions, turn over their focal adhesions and migrate over collagen surfaces compared to fibronectin-coated glass might depend on the engagement of α1β1 or α2β1 integrins, which favour collagen I. Perhaps these integrins depend less on branched actin for their dynamics or are present at lower levels than fibronectin receptors such as α5β1 or αVβ3, allowing dynamic turnover. The linear alignment of collagen fibers might also allow cells that cannot generate lamellipodia to persistently migrate or the stiffness of the substratum might affect the dynamics of migration. Clearly, more exploration of the mechanics behind the ability of *Nckap1* KO cells to migrate on collagen-coated surfaces needs to be done to uncover the mechanisms.

Much of the research on the requirement of the WRC during cell migration has been conducted in fast-moving amoeboid *Dictyostelium discoideum* [60], metastatic cancer cells such as A431 cells [11], mouse melanoma B16-F1 cells [31] or epithelial cancer cells [12]. Here we focused on mouse embryonic fibroblasts, and through inducible Cre-mediated recombination, we genetically deleted *Nckap1* resulting in the loss of the WRC (Figure 1). In the absence of lamellipodia, we observed pseudopod-like extensions in *Nckap1* KO cells that were more prevalent on collagen-coated surfaces, which correlated with increased cell migration (Figure 4). These pseudopod-like projects are akin to cells forming finger-like protrusions to migrate in soft 3D collagen matrices [49,61,62,63]. Computational modelling of these finger-like protrusions during migration suggests the cell would have various local actin networks, predominantly at the rear of the cell and highly enriched along the protrusion. Modelling would suggest a high level of actin-myosin stress within the protrusion, which decreases going back into the main body of the cell [64]. Indeed, we observed cells elongating and around the protrusion as the actin cytoskeleton straightens (Appendix A), which is likely driven by myosin mediated contractility to push and maintain the protrusion further and maintain a single elongated protrusion rather than many unsuccessful protrusions. Furthermore, it predicts that the cell produces a high level of traction forces at the protrusive tip [64], which we might expect from the *Nckap1* KO cells extending a pseudopod-like extension, anchoring to the matrix through an enrichment of focal adhesion proteins at the tip (Figure 3), and pulling the large body along when migrating over collagen (Figure 4 and Appendix A).

It will be interesting to know by what mechanisms WRC-depleted cells migrate more efficiently on collagen and whether this uses actin nucleated by formins [37,51] or other mechanisms such as microtubule-driven pseudopod growth [61]. It has recently been shown that mesenchymal cells migrating in 3D alter morphology, where lamellipodia are replaced by long pseudopod projections requiring a persistent assembly of microtubules to push and support long protrusions [65,66]. Moreover, high levels of myosin contractility and microtubules are required for rapid 3D migration through fibrillary matrices [67]. In the absence of the WRC (Figure 1 and Figure 2), we observed microtubules within and extending to the tips of these extensions of *Nckap1* KO MEFs, whereas the microtubules were largely absent from the lamellipodia region of control cells (Figure 1C). Therefore one hypothesis would be that there is cytoskeletal crosstalk, and microtubules may also promote pseudopod-like protrusions in the absence or dysregulation of the WRC during 2D cell migration. In normal migrating cells, microtubules are largely excluded from regions of actin-rich protrusions such as lamellipodia and filopodia [68] as they become unstable and buckle in areas under high retrograde flow such as the lamellipodia [69,70]. Slow-moving astrocytes form elongated protrusions that are enriched with microtubules with the protrusions. These cells have reduced actin dynamics that contain long stress fibers and well-developed focal adhesions running along the sides of the protrusion [68,71]. Therefore, one could envision that *Nckap1* KO cells, with their dysfunctional WRC, have reduced actin retrograde flows allowing microtubules to permeate the protrusion and deliver cargo and alter signalling events such as RhoG mediated disassembly of focal adhesion proteins [72]. The peripheral stress fibers and myosin contractility may act to keep the protrusion long-lived and inhibit any lateral extensions to keep the cell successfully migrating in one direction. Moreover, microtubules can interact with the actin cytoskeleton through formins such as mDia1 and mDia2, which work on the stabilisation of the microtubules [73,74,75]. Here, the microtubule end-binding protein 1 (EB-1), along with cytoplasmic linker protein 170 (CLIP-170) and mDia1, can accelerate the assembly of actin filaments from microtubule tips [76]. Furthermore, while focal adhesion turnover and cell migration are predominantly considered actin-based processes, there is increasing evidence that microtubules may have importance in these processes [77]. Microtubules are known to assist in focal adhesion dynamics [77,78,79,80], transport [81], and recycling [82]. Microtubules interact with focal adhesions through a complex involving CLASP, LL5β, Erc1, and liprin that link with Kank proteins and talin, which in turn are linking the actin cytoskeleton to integrins [77,83,84,85]. Therefore, it is possible that the WRC and actin dynamics predominate over 2D migration and that in the absence of the WRC and lack of N-WASP compensation, microtubules can take center stage, albeit at a non-optimal rate.

## Figures and Tables

**Figure 1 cells-09-01635-f001:**
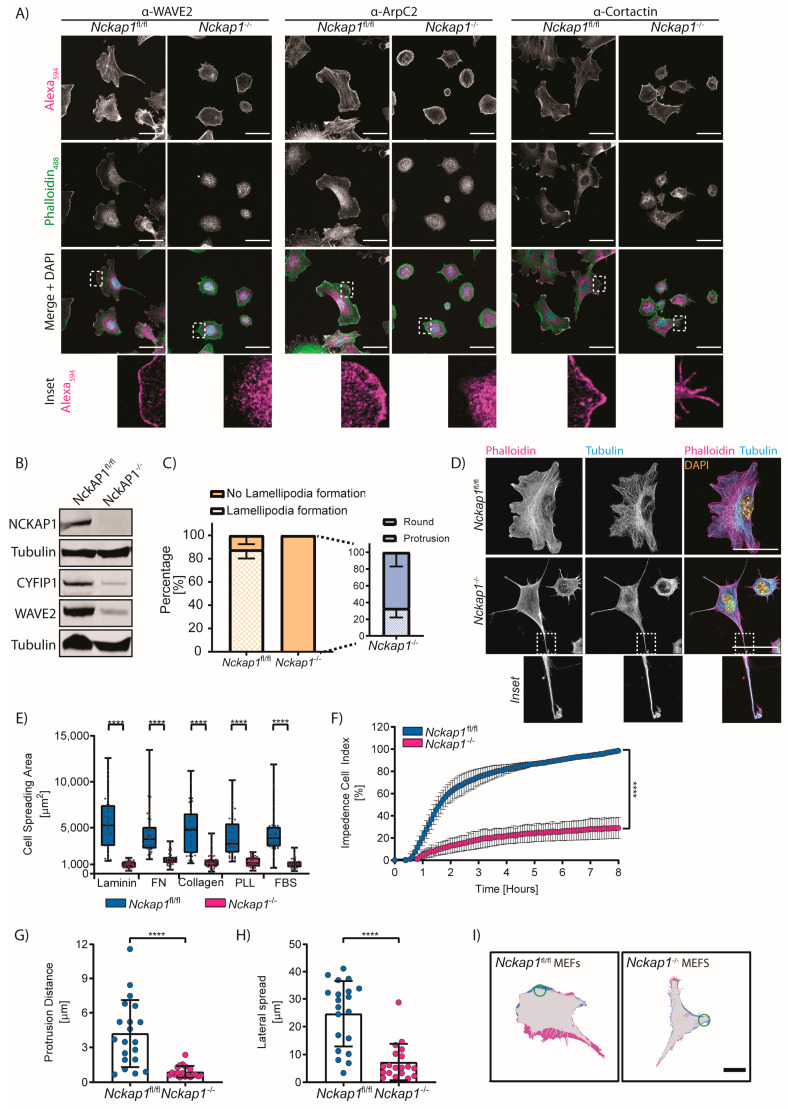
Effects on the lamellipodia in *Nckap1* KO cells. (**A**) Immunofluorescence analysis of lamellipodia components (WAVE2, ArpC2 or Cortactin) in *Nckap1* MEFs treated with either DMSO (control) or OHT (knockout). The actin cytoskeleton is shown by the phalloidin stain (green). The protein of interest is shown in magenta and an inset below showing the leading edge localisation of the protein of interest that is absent in the *Nckap1* KO cells. Scale bar = 50 μm. (**B**) Representative Western blots showing the loss of NCKAP1 and a significant down-regulation of other WRC components, WAVE2 and Cyfip1. Tubulin shown below protein tested relates to the loading control. *n* = 3 independent replicates. (**C**) Quantification of cells presenting with or without lamellipodia as a percentage. Inset represents *Nckap1* KO cells that do not present with lamellipodia, but rather with round or pseudopod-like protrusions as a percentage. Errors bars represent SD, *n* > 655 cells counted for each condition from 2 independent experiments. (**D**) Representative images of the shapes of these cells with phalloidin (magenta), α-Tubulin (cyan), and DAPI (yellow). Scale bar = 50 μm. (**E**) Spreading capabilities of control or *Nckap1* KO cells on various ECM-coated coverslips. Box and whisker plot with min, max, and mean represented. A 1-way ANOVA test between the control and *Nckap1* KO cells for each condition is shown. **** *p* < 0.0001. *n* = 2 independent experiments with 37 cells analysed in total. (**F**) The xCELLigence spreading assay where readings were taken every 10 min for 8 h. Data are normalised as a percentage of the control cell area at 8 h. Error bars represent S.E.M. from 3 independent experiments, which included 4 technical replicates per experiment. (**G**–**I**) Photoactivation of Rac1 on control and *Nckap1* KO cells. Protrusive distance after activation (**G**) and lateral spread of the protrusion after activation (**H**). Error bars represent S.D. *n* = >3 independent experiments from 20 cells. Statistical significance was assessed by a non-parametric Mann Whitney test **** *p* < 0.0001. (**I**) Endpoint overlay derived from the differences between time-points *t* = 30 (start of photoactivation) and *t* = 180 (end) depicting protrusions in cyan and retraction area in magenta. The area of photoactivation is shown in a green circle. Scale bar = 50 μm.

**Figure 2 cells-09-01635-f002:**
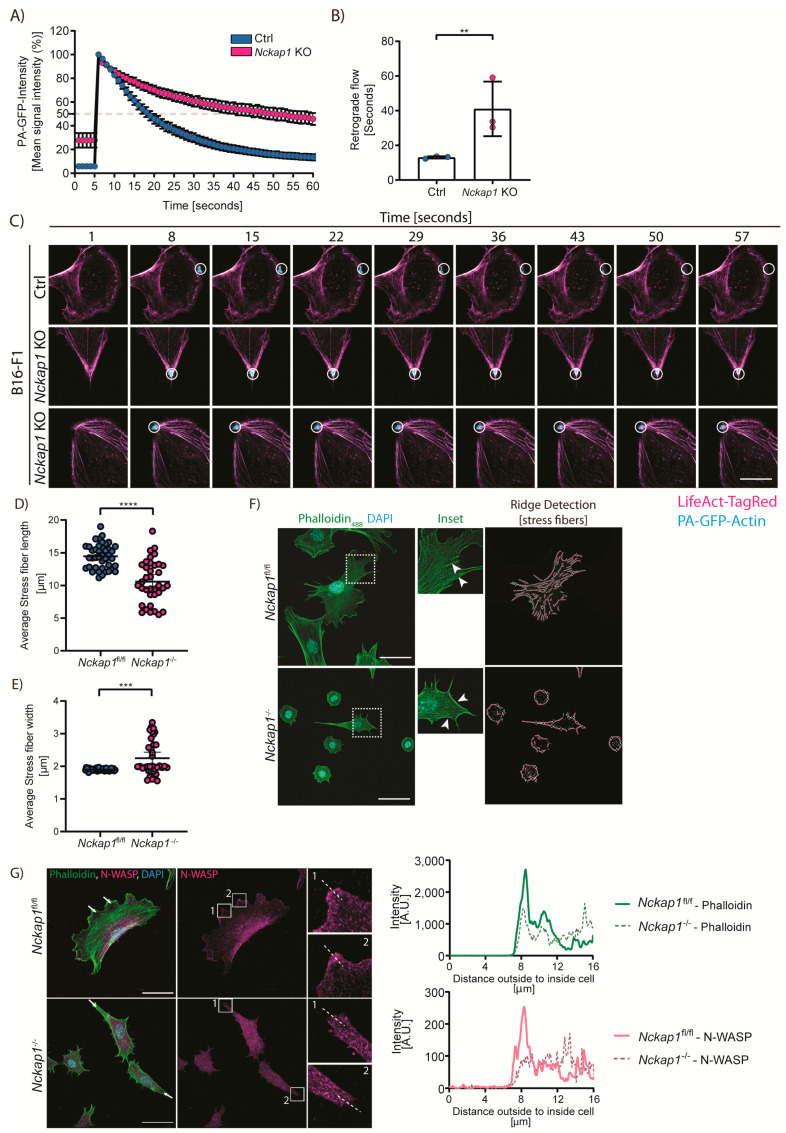
Deletion of *Nckap1* alters the actin dynamics. (**A**–**C**) Retrograde flow analysis in the control and *Nckap1* CrispR B16-F1 cells transfected with PA-GFP-Actin. (**A**) Quantification of PA-GFP-Actin signal intensity showing pre-bleach (the 5 s prior to activation), 100% (activation) and consequent intensity decay. The 60 cells were analysed over *n* = 3 independent experiments. Error bars represent 95% C.I. The dashed line is set at 50%. (**B**) Actin lifetime is expressed as *t*_1/2_ of intensity decay from (**A**). *n* = 3 independent experiments are shown. Error bars represent S.D. Statistical significance was assessed by a two-tailed *t*-test, ** *p* < 0.01. (**C**) Representative images showing PA-GFP-Actin (Cyan) and LifeAct-TagRed (Magenta) in B16-F1 (control) and *Nckap1* KO cells. The scale bar represents 50 μm. The supporting video is given in Appendix A. (**D**–**F**) Actin stress fiber quantifications from *Nckap1^fl/fl^* MEFs. (**D**) Stress fiber length and (**F**) Stress fiber thickness normalised to the cell area. (**D**,**E**) Error bars represent S.D. *n* > 37 images from 3 independent experiments. Statistical significance was assessed by a two-tailed *t*-test and a Mann-Whitney test for control and *Nckap1* KO cells, respectively. *** *p* < 0.001, **** *p* < 0.0001. (**F**) Representative cells stained with phalloidin and DAPI (left). The inset is shown focusing on stress fibers (middle); the white arrowheads highlight stress fibers. Actin stress fibers detected using ImageJ ridge detection overlaid on phalloidin image are shown (right). The scale bar represents 50 μm. (**G**) Endogenous localisation of N-WASP in control and *Nckap1* KO MEFs. MEFs plated on collagen I were fixed and stained with specific antibodies to F-actin (Green), N-WASP (Magenta), and DAPI (Cyan). The inset depicts a zoomed area used for the quantification of intensity for both Phalloidin and N-WASP. White arrows indicate the direction of the intensity quantification. The scale bar represents 50 μm, *n* = 2 cells of the control, and 5 cells from one experiment.

**Figure 3 cells-09-01635-f003:**
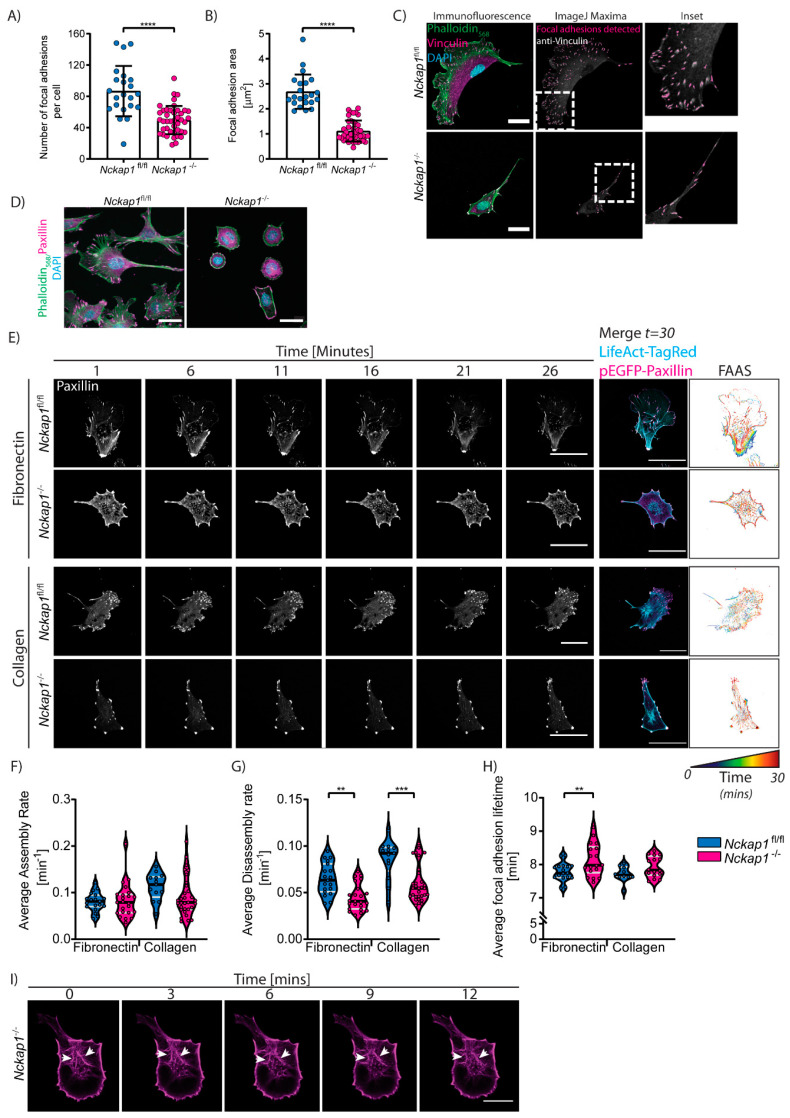
Loss of *Nckap1* affects focal adhesion dynamics. (**A**–**C**) Quantification of focal adhesions in control and *Nckap1* KO MEFs. (**A**) Quantification of the number of focal adhesions normalised to the cell area. (**B**) Quantification of focal adhesion area normalised to cell area. (**A**,**B**) *n*= 22 control and 42 *Nckap1* KO cells analysed from 2 independent experiments. The error bars represent S.D. Statistical significance was assessed using a two-tailed *t*-test with Welch’s correction (Focal adhesion numbers) and a non-parametric Mann Whitney test (Average adhesion area), **** *p* < 0.0001. (**C**) Representative images of control and *Nckap1* KO cells stained with Phalloidin, Vinculin (focal adhesion marker) and DAPI. Focal adhesion parameters (3A-B) were quantified using the ImageJ maxima tool. The output is shown in magenta overlaid on the original vinculin channel to display the robustness of the detection. Scale bar represents 50 μm. (**D**) Representative images of control and *Nckap1* KO cells stained with Phalloidin, Paxillin (another focal adhesion marker), and DAPI. (**E**–**G**) Control and *Nckap1* KO cells expressing LifeAct-TagRed and pEGFP-Paxillin spreading and migrating on fibronectin or collagen. Images were taken every minute for a total of 30 min. The supporting video is given in Appendix A. Focal adhesion turnover was assessed from the focal adhesion analysis server (FAAS). (**E**) Representative images of focal adhesion changes over time. The right-hand panels represent the merge of the actin cytoskeleton and paxillin from a single time point next to the FAAS output turnover image. The scale represents the changes to the focal adhesions from blue to red over time. The scale bars represent 50 μm. (**F**–**H**) Focal adhesion analysis server output of assembly rates (**F**), disassembly rates (**G**), and adhesion lifetime (**H**) between control and *Nckap1* KO cells on either FN or collagen. Violin plots display the median (black) and quartiles (grey), *n* > 13 cells analysed from 3 independent experiments. Statistical significance was assessed using a 1-way *ANOVA*, ** *p* < 0.01, *** *p* < 0.001. (**I**) Time-course highlighting the dynamic actin cage surrounding the nucleus of *Nckap1* KO MEFs. White arrowheads indicate thick actin fibers around the nucleus. Time is given in minutes, and the scale bar represents 20 μm.

**Figure 4 cells-09-01635-f004:**
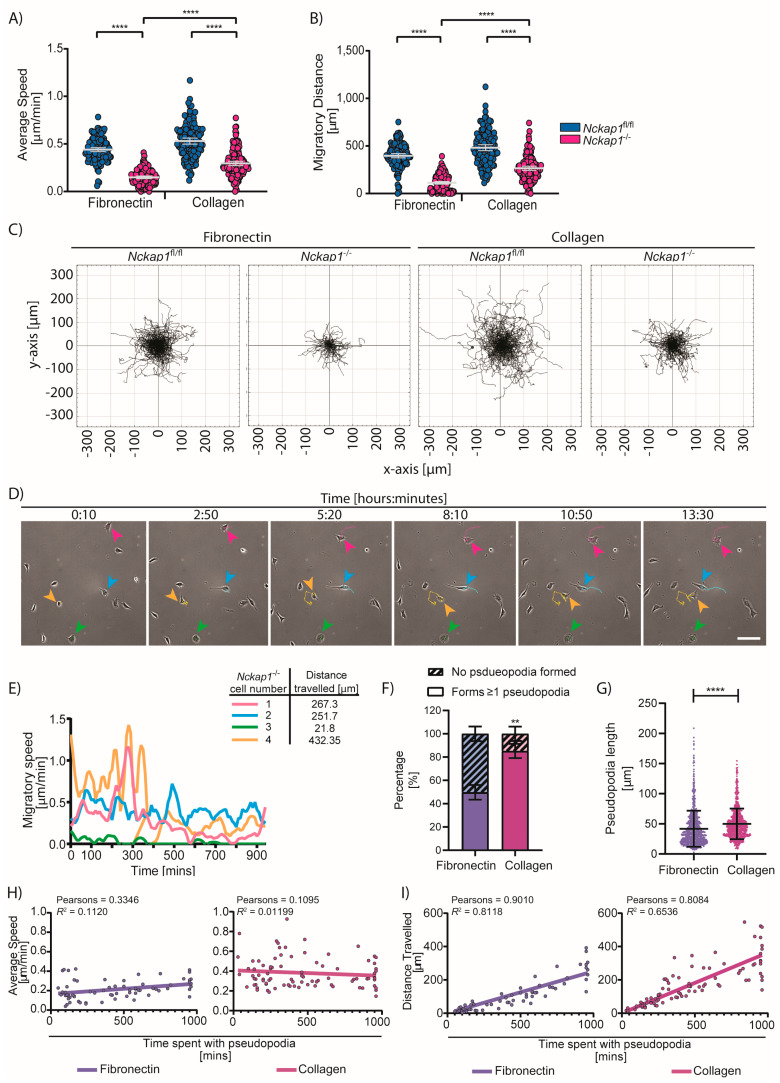
Collagen can partially rescue the migratory defect in *Nckap1* KO cells. The random migration assay of the control and *Nckap1* KO cells migrating over fibronectin and collagen is shown. (**A**) Migratory distance travelled, (**B**) average migratory speed. The dot plots present mean and S.D. from >130 cells analysed for each condition from 2 independent experiments. Statistical significance was assessed using a 1-way *ANOVA* Kruskal-Wallis test, **** *p* < 0.0001. (**C**) Spider plots represent migratory paths and distance travelled. (**D**,**E**) Representative migratory profiles of four *Nckap1* KO cells that represent rounded, pseudopod-like protrusions and cells that transition between the two phenotypes. Note: The green cell is rounded for the entire experiment and has very little speed or distance travelled. The cyan cell forms a pseudopod-like extension early and migrates at a steady speed throughout the experiment. Cells 3 and 4 (green and orange) transition between states and fast and slow bursts of speed over time that correlate with their shape. (**D**) Still images displaying a time-course of *Nckap1* KO MEFs migrating over collagen. Time is shown as hours:minutes, and the scale bar represents 100 μm. Arrows pointing to the cell in question and its migration path are shown; see supporting video in Appendix A. (**E**) Cell migration speed over time for the four cells in (**D**) analysed using the ImageJ MTrackJ plugin. Coloured lines on graph match the cells in (**D**). The migratory distance for each cell is also depicted. *n* = 4 cells from one video. (**F**) Percentage of *Nckap1* KO cells that form pseudopod-like extensions within the 16 h experiment on either fibronectin or collagen. A total of 107 cells (fibronectin) and 93 cells (collagen) were analsyed from 3 movies each. The error bars represent S.D. Statistical significance was assessed using a two-tailed *t*-test with Welch’s correction, ** *p* < 0.01. (**G**) The length of these pseudopod-like extensions was determined from cells forming a protrusion every hour in a 16 h movie. A total of 854 (Fibronectin) and 862 (collagen) pseudopod-like extensions were measured. Error bars represent S.D. The error bars represent S.D Statistical significance was assessed using a two-tailed *t*-test with Welch’s correction, **** *p* < 0.0001. (**H**,**I**) Correlations between lifetimes of the pseudopod extensions with either speed (**H**) or migratory distance (**I**). A total of 68 cells (fibronectin) and 86 cells (collagen) were analsyed from 3 movies each. Pearson’s coefficient and *R*^2^ value are shown for each condition.

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
