# Peer review of "The WAVE Regulatory Complex Is Required to Balance Protrusion and Adhesion in Migration"

_cells, 2020, doi:10.3390/cells9071635_

Round 1

Reviewer 1 Report

The manuscript by Whitelaw, Swaminathan & Machesky explores the role of the Wave regulatory complex component, NckAP1, in cell migration. The individual contributions of actin regulatory protein complexes (Arp2/3, scar/WAVE, fascin, Mena/VASP) to different actin architectures and resulting cellular structures, is an area of intensive research. Despite this, the link between WRC-dependent branched actin dynamics, lamellipodial formation, and cell-matrix adhesion, is not well understood. This manuscript builds from observations that the WRC is required for focal-adhesion signalling, turnover and integrin recycling. To perturb the WRC, they utilise NckAP1 depleted MEFs, and B16-f1 mouse melanoma CRISPR KO lines, and perform a logical analysis of cell migration investigating signalling, actin dynamics and cell-matrix adhesion dynamics. They nicely tie in their work to our current understanding of other studies investigating other actin regulators in these functions. Although not profound, the authors aptly acknowledge concepts their study does not address, and present a sound story with clean data.

I commend the authors on presenting colour-blind friendly, and greyscale images, in addition to scatterplots and violin plots, which display the spread of data.

I suggest the following minor comments/edits:

My largest concern is the lack of 3D data. I would strongly suggest the authors compare control vs CRISPR B16 cells embedded in collagen, polymerised at 37C vs 22C. The Machesky lab has previously published a circular invasion assay, which would also suit this purpose. This would address if the cells are switching to invasive pseudopodia in collagen i.e. are these more invasive once embedded in 3D? The two collagen gelation protocols might address biology pertaining to mechanics and topography of the microenvironment (see Yamada lab’s work on CDMs vs collagen). Furthermore, large adhesive structures analogous to the focal adhesions we observe in 2D are not as prevalent in 3D migration. Evidence is emerging that the actin structures are also somewhat different. The manuscript discussion would benefit from a sentence or two addressing these differences.

The discussion could also be strengthened by the addition of commentary around microtubules growing into areas of high retrograde flow. Clare Waterman’s early work (10.1083/jcb.139.2.417) demonstrates microtubules buckle in areas of high retrograde flow (i.e. leading edge of cells). Potentially, as retrograde flow is lessened in the Nckap1 KO cells, microtubules may be able to enter cell extensions. As microtubules also target cargo (i.e. Rho GEFs), this may affect that signalling. 

They report a defect in adhesion turnover, measure the assembly and disassembly rates. Does this affect the adhesion life-time? This should be added to figure 3.

Line 350. The authors refer to a strong cage-like actin network around the nucleus of these cells (Fig 3e). This data needs to be presented more convincingly in the figure, it is apparent in the supplementary movie. Arrow heads and magnified regions would aid in visualising this as it may be missed by the untrained eye.

Are the Nckap1 KO MEFs senescent?

I found the first paragraph oddly placed- all data listed was supplementary. Perhaps the immunoblot could be moved to the first figure.

Typo Figure 3a. The graph Y axis should read “number of FA per cell” (written currently, “per to cell”)

Supplementary movie 2, this would benefit from labeling.

Author Response

1) My largest concern is the lack of 3D data. I would strongly suggest the authors compare control vs CRISPR B16 cells embedded in collagen, polymerised at 37C vs 22C. The Machesky lab has previously published a circular invasion assay, which would also suit this purpose. This would address if the cells are switching to invasive pseudopodia in collagen i.e. are these more invasive once embedded in 3D? The two collagen gelation protocols might address biology pertaining to mechanics and topography of the microenvironment (see Yamada lab’s work on CDMs vs collagen).

While we agree that it could be interesting to explore 3D migration in a collagen plug, this opens a whole new area of research and with the COVID-19 lockdown restrictions we do not have the access needed to the lab to perform these experiments.

2) Furthermore, large adhesive structures analogous to the focal adhesions we observe in 2D are not as prevalent in 3D migration. Evidence is emerging that the actin structures are also somewhat different. The manuscript discussion would benefit from a sentence or two addressing these differences.

We agree with the referee’s suggestion and we have added a few sentences around FAs in 2D and 3D and a new reference addressing this. Namely, the sentence starting on Line 538: “While there are parallels between Nckap1 KO cells migrating in 2D and cells migrating in 3D collagen matrices is nice, there are also some differences in structures such focal adhesions. Notably, focal adhesions tend to be smaller and distributed over the surface of the cell in contact with the matrix, rather than the large elongated or arrow-head shaped structures commonly observed in 2D on rigid substrates. Additionally, focal adhesion proteins in 3D regulate cell speed through matrix deformation [54].”

3) The discussion could also be strengthened by the addition of commentary around microtubules growing into areas of high retrograde flow. Clare Waterman’s early work (10.1083/jcb.139.2.417) demonstrates microtubules buckle in areas of high retrograde flow (i.e. leading edge of cells). Potentially, as retrograde flow is lessened in the Nckap1 KO cells, microtubules may be able to enter cell extensions. As microtubules also target cargo (i.e. Rho GEFs), this may affect that signalling.

We added a few sentences to the discussion and some new references, starting on Line 598:
In normal migrating cells microtubules are largely excluded from regions of actin rich protrusions such as the lamellipodia and filopodia [67] as they become unstable and buckle in areas under high retrograde flow such as the lamellipodia [68, 69]. Slow moving astrocytes form elongated protrusions that are enriched with microtubules with the protrusions. These cells have reduced actin dynamics that contain long stress fibers and well developed focal adhesions running along the sides of the protrusion [67, 70]. Therefore, one could envision that Nckap1 KO cells, with their dysfunctional WRC, have reduced actin retrograde flows allowing microtubules to permeate the protrusion and deliver cargo and alter signalling events such as RhoG mediated disassembly of focal adhesion proteins [71]. The peripheral stress fibers and myosin contractility may act to keep the protrusion long- lived and inhibit any lateral extensions to keep the cell successfully migrating in one direction.”

4) They report a defect in adhesion turnover, measure the assembly and disassembly rates. Does this affect the adhesion life-time? This should be added to figure 3.

We agree that this is interesting and we measured the adhesion life-time using the FAAS outputs and added this into the figure. Adhesion lifetime is significantly longer in Nckap1 KO cells on FN. Now Figure 3H.

5) Line 350. The authors refer to a strong cage-like actin network around the nucleus of these cells (Fig 3e). This data needs to be presented more convincingly in the figure, it is apparent in the supplementary movie. Arrow heads and magnified regions would aid in visualising this as it may be missed by the untrained eye.

We now show an image of the actin-cage around nucleus as suggested, taken from stills. This is now as figure 3I with white arrow heads indicating the stable actin cables around the nucleus.

6) Are the Nckap1 KO MEFs senescent?

This is possible, but it is beyond the scope of the study and as we haven’t tested for senescence in the MEFs, we haven’t discussed this.

7) I found the first paragraph oddly placed- all data listed was supplementary. Perhaps the immunoblot could be moved to the first figure.

The immunoblot is now moved from the supplementary to figure 1b.

8) Typo Figure 3a. The graph Y axis should read “number of FA per cell” (written currently, “per to cell”)

This is now changed in the figure 3

9) Supplementary movie 2, this would benefit from labeling.

All movies have been updated with labels and time stamps.

Reviewer 2 Report

In this manuscript, Whitelaw et al., assess the effect of Nap1 (a WAVE regulatory complex component) KO on the actin cytoskeleton, focal adhesions and cell migration in 2D. The major strengths of the study are the various quantitative assessments on the form and dynamics of the actin cytoskeleton and focal adhesions (KO vs WT), as well as good presentation of the data in the figures.

Major comments:

This part of the abstract and the corresponding parts of the main text do not seem to be fully supported by evidence: “In the absence of lamellipodia, cells migrating on collagen can become elongated and move with a single thin pseudopod, which appears devoid of N-WASP. Thus, cell migration on collagen is less dependent on branched actin than migration across flat 2D surfaces.”

Firstly, the migration on collagen that the authors assess (in this paper) appears to be on collagen (col) coated 2D surface, so the contrast drawn with 2D surfaces (in the quoted last sentence above) seems inappropriate. More importantly, the WT cells themselves migrate better on col than fibronectin (Fn). The KO cells also show a similar trend. The KO cells on Fn also appear to form pseudopods, so the claim of pseudopod-based migration being something specific on col appears an overreach.

The title is vague and more apt for a review article. Should be changed to more accurately reflect their actual findings.

The motivation for the work is not clear from the introduction. Given that the effect of Nap1 depletion (including some effects on focal adhesions) was studied in Tang et al Curr. Biol. 2013, what was the specific motivation for this work? The focus on a different cell or more/different molecular details? This can be better expressed in the Introduction.

Minor comments:

Line 207: “…cortactin localised to filopodia…” is the distinction being made between bonafide filopodia and filopodia-like structures? Which one is this?

There are quite a few language errors: eg. “While, WAVE knockdown cells displayed larger focal adhesions connected to actin stress fibers.”,  “NckAP1 belongs to the Hem family of transmembrane protein[s]…”, “…cell spreading continually in real-time using [28].”, “Using antibodies specific, we analysed the endogenous localisation of N-Wasp…”, “in in the absence of the WRC”

Fig 3A y axis label: “…per to cell” should be “per cell”?

Line 452: “… the increased ability of the NckAP1 KO cells to form the pseudopod-like protrusions, turn over their focal adhesions and migrate over collagen surfaces” – should make it clear that this is in comparison to the KO cells on fibronectin surfaces (as opposed to: in comparison to WT cells on collagen surfaces)

Similarly, Line 461: “In the absence of a lamellipodia, we observed pseudopod-like extensions that correlated with increased cell migration” – again, should make it clear that this is in comparison of the KO cells on col vs Fn (as opposed to: KO vs WT cells)

Line 456: alpha5beta1 is mentioned as the fibronectin receptor; would, say, alphaVbeta3 not be relevant?

References appear to be cited in an inconsistent manner: eg., “…demonstrated for the Scar/WAVE protein alone [4] and later for several subunits of the WRC (Steffen et al. 2004).”

Author Response

Major comments:

This part of the abstract and the corresponding parts of the main text do not seem to be fully supported by evidence: “In the absence of lamellipodia, cells migrating on collagen can become elongated and move with a single thin pseudopod, which appears devoid of N- WASP. Thus, cell migration on collagen is less dependent on branched actin than migration across flat 2D surfaces.”

1) Firstly, the migration on collagen that the authors assess (in this paper) appears to be on collagen (col) coated 2D surface, so the contrast drawn with 2D surfaces (in the quoted last sentence above) seems inappropriate.

We updated the end of the abstract starting on Line 20 with:

“In the absence of lamellipodia, cells can become elongated and move with a single thin pseudopod, which appears devoid of N-WASP. This phenotype was more prevalent on collagen than fibronectin where we observed an increase in migratory speed. Thus, 2D cell migration on collagen is less dependent on branched actin.”

2) More importantly, the WT cells themselves migrate better on col than fibronectin (Fn). The KO cells also show a similar trend.

We mention in the text that both migrate better on collagen but the Nckap1 KO cells have a greater rescue on collagen.
We calculated the differences between WT FN to WT Coll is an increase in ~17% whereas KO FN to KO Coll is an increase in ~49 % of speed. – Not shown in the text

Namely this sentence starting on Line 474:
“This ability of the Nckap1 KO cells to polarize and form a protrusion and thus migrate is observed on both fibronectin and collagen coated surfaces, however this migratory mode was more prevalent on collagen-coated surfaces where we observe an increase in speed also in the distance travelled.”

3) The KO cells on Fn also appear to form pseudopods, so the claim of pseudopod-based migration being something specific on col appears an overreach.

We have now updated the manuscript with the sentence to indicate that more cells successfully form a protrusion on collagen than FN starting on Line 474.

This ability of the Nckap1 KO cells to polarize and form a protrusion and thus migrate is observed on both fibronectin and collagen coated surfaces, however this migratory mode was more prevalent on collagen coated surfaces where we observe an increase in speed and also in the distance travelled.

4) The title is vague and more apt for a review article. Should be changed to more accurately reflect their actual findings.

Line 2: changed to “The WAVE regulatory complex is required to balance protrusion and adhesion in migration.”

5) The motivation for the work is not clear from the introduction. Given that the effect of Nap1 depletion (including some effects on focal adhesions) was studied in Tang et al Curr. Biol. 2013, what was the specific motivation for this work? The focus on a different cell or more/different molecular details? This can be better expressed in the Introduction.

We have now updated the introduction with the sentences beginning Line 61:

“There is a significant body of evidence describing the functions of WRC complex. However, these systems do not faithfully reflect the normal physiological functions of WRC complex. In vivo functional studies using mouse model is hampered by prenatal lethality phenotypes of

WRC complex members. Therefore, we established an inducible Nckap1 floxed mouse model to robustly isolate MEF for WRC functional studies.”

Minor comments:

6) Line 207: “...cortactin localised to filopodia...” is the distinction being made between bonafide filopodia and filopodia-like structures? Which one is this?

It was filopodia-like structures. Now changed at Line 265:

We did not observe many filopodia-like structures in control MEFs whereas Nckap1 KO MEFs (Nckap1-/-) had numerous filopodia-like structures where we observed cortactin localising (Figure 1a), similar to what has been shown for filopodia in growth cones [31].”

7) There are quite a few language errors: eg. “While, WAVE knockdown cells displayed larger focal adhesions connected to actin stress fibers.”,

Line 57 now changed to: “Silva et al. found that WRC knockdown cells displayed larger focal adhesions connected to actin stress fibers”

NckAP1 belongs to the Hem family of transmembrane protein[s]...”,

Now changed at Line 65: “NCKAP1 belongs to the HEM family of proteins, originally thought to be transmembrane proteins, but now known to be cytoplasmic and is conserved as a subunit of the WRC in a wide range of organisms.“

“...cell spreading continually in real-time using [28].”,
Now changed at Line 286: “We also used xCELLigence (ACEA Biosciences Inc.) to measure

cell spreading continually in real-time [34]”

“Using antibodies specific, we analysed the endogenous localisation of N-Wasp...”,
Line 364
now changed to: “Using a specific N-WASP antibody, we analysed the endogenous

localisation of N-WASP in Nckap1 KO cells that have a pseudopod-like protrusion” “in in the absence of the WRC”

Line 531 now removed the 2nd “in” and now changed to: Recently it was shown in Dictyostelium discoideum that active Rac1 was able to drive pseudopod extensions even in the absence of the WRC [41].”

Fig 3A y axis label: “...per to cell” should be “per cell”?

We have now changed this in the figure

8) Line 452: “... the increased ability of the NckAP1 KO cells to form the pseudopod-like protrusions, turn over their focal adhesions and migrate over collagen surfaces” – should make it clear that this is in comparison to the KO

cells on fibronectin surfaces (as opposed to: in comparison to WT cells on collagen surfaces)

Sentence changed beginning on line 558 to:

We hypothesise that the increased ability of the NckAP1 KO cells to form the pseudopod-like protrusions, turn over their focal adhesions and migrate over collagen surfaces compared to fibronectin might depend on engagement of α1β1 or α2β1 integrins, which favour collagen I.”

9) Similarly, Line 461: “In the absence of a lamellipodia, we observed pseudopod- like extensions that correlated with increased cell migration” – again, should make it clear that this is in comparison of the KO cells on col vs Fn (as opposed to: KO vs WT cells)

Sentence now changed to on Line 572:

In the absence of a lamellipodia, we observed pseudopod-like extensions Nckap1 KO cells that were more prevalent on collagen coated surfaces, which correlated with increased cell migration”

10)Line 456: alpha5beta1 is mentioned as the fibronectin receptor; would, say, alphaVbeta3 not be relevant?

These are both FN receptors and α5β1 as mentioned is the most abundant FN receptor but we changed the sentence to on Line 561:

“Perhaps these integrins depend less on branched actin for their dynamics or are present at lower levels than fibronectin receptors such as α5β1 or αVβ3, allowing dynamic turnover.”

11)References appear to be cited in an inconsistent manner: eg., “...demonstrated for the Scar/WAVE protein alone [4] and later for several subunits of the WRC (Steffen et al. 2004).”

Changed within text to numbered

Reviewer 3 Report

The manuscript by Whitelaw et al describes the morphological and migratory phenotypes of cultured cells depleted of NckAP1. Depletion of nckap1 results in rounded or pseudopod-containing cells that show loss of normal actin, branch actin nucleating proteins, and focal adhesion proteins. The cells show abnormal movement by a number of assays on a variety of substrates. The data is of high quality, the statistics are sound, and the interpretations are logical. I wish the journal would have provided high resolution images, as it is hard to see many of the structures in the compressed PDF images. I have only minor comments/questions/suggestions, mostly on methodologies.

Can the authors speculate on why there appear to be two separate phenotypes (small and round vs. angular and pointy)? It seems that the rounded cells represent a more severe phenotype. Was there any change over time in the ratio of these two? Note there is a binucleate rounded cell in video 3, and another cell dies during the imaging. Are these common occurrences?

Looking at video 2, the stress fibers surrounding the cell periphery are connected between protrusion points and often start off bent, but then straighten out, making the cell boundary locally straighter and overall more angular. It is almost as if the fibers are slid past one another to straighten. Can the authors speculate on this mechanism? Myosins?

What is the reason for the huge accumulations of FAs at the pointy tips of the angular nckap1 mutant cells (seen clearly in figure 3E)?

Figure 1C, it is stated that MTs are enriched in the branches in the nckap1-depleted cells, but the image is low magnification and hard to see. A zoom and/or quantification may help.

Figure 1 legend, POI not defined (I think protein of interest though).

Figure 1H, it’s not clear how the image was made.

Figure 2C: it would help to put a box at the ROIs used for photoactivation. It’s more obvious in the video (video 1). The scale bars on every image with tiny text is not the usual convention. Usually just one bar. The control shows most of the cell, but the mutant shows only part, and without reference to the cell’s axis. It would be nice to see the rest of it. In the control, is there a reason why PA GFP-actin is also showing up at the rear of the cell even prior to photoactivation?

Figure 2D-F:  only the rounded nckap1-depleted phenotype is shown, and it’s hard to discern stress fibers in the images of them. It may be helpful to put a higher resolution image for these. The imageJ ridge detection plugin has a lot of parameters that can lead to very different outputs. Were the same parameters used for comparison, and were all the data in figure 2D derived from ridge detection? Normalizing for cell area for D and E is logical, but for stress fiber thickness (2E), would this be necessary?

Figure 2G, what is the reason for the 10-fold fluorescence intensity differences on the y-axes of the intensity plots?

Is the TagRed protein TagRFP?

All the videos show panel borders on the first frame only. It would be better to remove them altogether or keep on all frames. Just a stylistic suggestion though.

Figure 3C:  the imageJ-derived maxima are overlaid onto the gray image. Being unfamiliar with the plugin, I’m not sure what this is showing. Is the signal intensity proportional to anything? If not, then maybe it’s not necessary. It looks nice though. Clarification would be helpful.

Figure 3E:  is the frame with the actin/paxillin overlay a time projection? The FAAS output is fancy, but could benefit from a bit more explanation. And it would help to have the time ramp at the bottom overlaid with the colormap used for the images. The Figure 3A and C legend text is reversed.

Supplemental videos should have more detail in the legends. At least scale bar sizes and units for the times. Time units on the video itself ideally.

Video 2:  the four panels should be labeled. There is a lot of flicker in some of the cells. Was auto-contrast used for each frame? Time registration to fix the slight drift of the cells would help the quality and ease of viewing. The upper left cell is beautiful, but it is too bad there are all the focal plane changes. OK to leave, but personally this reviewer would show a better one if possible.

Minor stuff:

Line 60:  “is conserved”

Line 81:  spelling for hydroxytamoxifen

Line 86:  include primer sequences.

Line 109 is a different font.

Line 148:  define FAAS after its full name since line 151 uses the abbreviation

Line 163:  “images were processed”

Line 173:  use µm2 for stating ROI size instead of pixel count.

Line 191:  Put the full method instead of citing a paper that is not even published yet, which is cited as having been previously described. Always err on the side of detailed methods and never (within reason) just cite another paper. Everybody hates that, but yet everybody has done it (present reviewer included regrettably).

Line 202-203:  reference format inconsistent

Line 207:  comma after MEFS

Line 226:  using [28]? Put the name in front of the reference.

Line 254:  “ECM-coated”

Line 285:  “…KO cells is reminiscent…”

Line 296-297:  grammar

Line 312:  “experiments”

Line 340:  “to have an impact on focal…” or “to impact focal…”

Figure 3A:  y-axis grammar

Figure 3C label text has a space, reads as “Immunof luorescence"

Figure 3C label text there is a space in detected, reads as “det ected”

Line 395:  says violin plots present median, but there does not appear to be any indication of this on the plots.

Figure 4A and B, there are significance bars for between-cell comparison, but the text mentions also a significant difference for the between-substrate comparison. Can this be presented as well? Given that the figure title is that collagen partly rescues the migratory defect, it seems this may be a relevant detail to show.

Line 422:  Amato et al, inconsistent reference style

Line 460:  comma after fibroblasts (unless you want the reader to read the sentence 3 times to figure it out.)

Line 474:  period after sentence missing

Author Response

1) Can the authors speculate on why there appear to be two separate phenotypes (small and round vs. angular and pointy)? It seems that the rounded cells represent a more severe phenotype. Was there any change over time in the ratio of these two?

We believe that the rounded cells with some points transition to form one point/protrusions and migrate. We added in the sentences (Line 466 and 483) below to the text and also analysed 4 cells in the random migration assay to show the transition between the two phenotypes (Now figure 4d,e):

(Line 466) We next analysed whether Nckap1 KO cells migrating over collagen transitioned between the two phenotypic shapes described in Figure 1c and if this affected cell migration speeds. We observed that the majority of Nckap1 KO cells remained round and barely moved throughout the experiment. Rounded cells remain relatively immobile and correlated with reduced speeds and migratory distance (Figure 4d,e; S. Video 4). Nckap1 KO cells that maintain a pseudopod-like extension for migration appear to maintain a relatively steady speed throughout the experiment (Figure 4d,e; S. Video 4). Few cells were able to transition from a rounded shape to form a pseudopod-like protrusion and were able to increase their speeds and migrate further (Figure 4e; S. Video 4).

Line 483: “We speculate that the actin stress fibers and slower focal adhesion turnover results in a strong actin cytoskeleton that has to be overcome to transition from the rounded to a pseudopod-like protrusive phenotype for migration.”

2) Note there is a binucleate rounded cell in video 3, and another cell dies during the imaging. Are these common occurrences?

We do observe binucleated KO cells, possible due to defects in the cell cycle which was not tested.
Cells are plated for 4 hours prior to imaging, in this time many KO cells do not attach properly and some will die throughout the experiments

3) Looking at video 2, the stress fibers surrounding the cell periphery are connected between protrusion points and often start off bent, but then straighten out, making the cell boundary locally straighter and overall more angular. It is almost as if the fibers are slid past one another to straighten. Can the authors speculate on this mechanism? Myosins?

We mentioned this in the discussion
Line 578: “Modelling would suggest a high level of actin-myosin stress within the protrusion, which decreases going back into the main body of the cell.”
And added in this sentence after:
Line 579: “Indeed, we observed cells elongating and around the protrusion as the actin cytoskeleton straightens (S. Video 2), which is likely driven by myosin mediated contractility to push and maintain the protrusion further and maintain a single elongated protrusion rather than many unsuccessful protrusions.”

4) What is the reason for the huge accumulations of FAs at the pointy tips of the angular nckap1 mutant cells (seen clearly in figure 3E)?

We are unsure but we added in a sentence:

Line 416: “We speculate that the accumulation of focal adhesion proteins at the tips of the protrusions may help in anchoring the front of the protrusion during migration.”

5) Figure 1C, it is stated that MTs are enriched in the branches in the nckap1- depleted cells, but the image is low magnification and hard to see. A zoom and/or quantification may help.

A zoom has been added to figure 1D to show microtubule in the branches

6) Figure 1 legend, POI not defined (I think protein of interest though).

Line 311 Changed to “Protein of interest”
7) Figure 1H, it’s not clear how the image was made.

We have now explained this in Figure 1 legend and the methods sections

Line 229: “Endpoint overlay derived from the differences between time-points t=30 (start of photoactivation) and t=180 (end) depicting protrusions in cyan and retraction area in magenta.”

8) Figure 2C: it would help to put a box at the ROIs used for photoactivation. It’s more obvious in the video (video 1).

ROIs have been added for both stills and S. Movie 1

9) The scale bars on every image with tiny text is not the usual convention. Usually just one bar.

Now only one singe scale bar without text

10)The control shows most of the cell, but the mutant shows only part, and without reference to the cell’s axis. It would be nice to see the rest of it.

It is because of the objective used, we don’t get all of the cell in the field of view and had to focus on a region. I have also added a second Nckap1 KO image that has much more of the cell within the field of view.

11)In the control, is there a reason why PA GFP-actin is also showing up at the rear of the cell even prior to photoactivation?

This is likely a little channel bleed-through between channels as the images are acquired by line and not sequentially for increased acquisition speeds of 2 colour and photoactivation.

12)Figure 2D-F: only the rounded nckap1-depleted phenotype is shown, and it’s hard to discern stress fibers in the images of them. It may be helpful to put a higher resolution image for these.

We added a new image for figure 2F with both rounded and pseudopod-like extensions shown.
We also added an inset focusing on a stress fiber region and arrows to highlight these.

13) The imageJ ridge detection plugin has a lot of parameters that can lead to very different outputs. Were the same parameters used for comparison, and were all the data in figure 2D derived from ridge detection?

We have added the parameters used to detect the stress fibers from the Ridge detection into the methods section. The same parameters were used in both WT and KO cells. Only difference is that we manually remove the leading edge/lamellipodia actin as this is now stress fiber actin. Described in the methods starting at Line 199.

14)Normalizing for cell area for D and E is logical, but for stress fiber thickness (2E), would this be necessary?

We believe that normalising to cell area justified since there is a drastic difference between control and KO cell area

15)Figure 2G, what is the reason for the 10-fold fluorescence intensity differences on the y-axes of the intensity plots?

These are intensity plots of two different antibodies; Phallodin (actin) vs N-WASP. Solid line is WT and dashed line is KO

16)Is the TagRed protein TagRFP?

It is a variant of RFP, the construct is known as LifeAct-TagRed as described in Brooks et al., (2010) The Nance-Horan syndrome proteins encodes a functional WAVE homology domain (WHD) and is important for coordinating actin remodelling and maintaining cell morphology.

17)All the videos show panel borders on the first frame only. It would be better to remove them altogether or keep on all frames. Just a stylistic suggestion though.

All the videos have been changed. Borders have been added to each frame and labels

18)Figure 3C: the imageJ-derived maxima are overlaid onto the gray image. Being unfamiliar with the plugin, I’m not sure what this is showing. Is the signal intensity proportional to anything? If not, then maybe it’s not necessary. It looks nice though. Clarification would be helpful.

We added a sentence to the methods section starting at Line 171:
The output image from the ImageJ-derived maxima is overlaid onto a greyscale image of the FAs from the original file to indicate that the method is able to distinguish the majority of focal adhesion proteins from the original image

19)Figure 3E: is the frame with the actin/paxillin overlay a time projection?

This comes from a single frame from one time point (t=30 mins) An explanation was added to the legend see Line 447

20)The FAAS output is fancy, but could benefit from a bit more explanation. And it would help to have the time ramp at the bottom overlaid with the colormap used for the images. The Figure 3A and C legend text is reversed.

Added more details in the methods section:

Line 183: “The server is set up to calculate assembly and disassembly rates using the same method as described in [24] with the modifications described in [25]. Briefly, the software follows the mean intensity of Paxillin through time and fits a linear model to the log- transformation time series of the Paxillin intensity as described in [25]. FAAS also calculates dynamic properties such as focal adhesion longevity (lifetime) from the birth and death of focal adhesions between frames described in [25]. “

Added the time in the colourmap

And changed figures 3a and C around.

21)Supplemental videos should have more detail in the legends. At least scale bar sizes and units for the times. Time units on the video itself ideally.

We have added legends for all supplementary movies. See lines 673-694

22)Video 2: the four panels should be labeled. There is a lot of flicker in some of the cells. Was auto-contrast used for each frame? Time registration to fix the slight drift of the cells would help the quality and ease of viewing. The upper left cell is beautiful, but it is too bad there are all the focal plane changes. OK to leave, but personally this reviewer would show a better one if possible.

We have labelled the panels now
Minor stuff:
23) Line 60: “is conserved”
Line 67
: are is changed to “is”
24) Line 81: spelling for hydroxytamoxifen changed on Line 104

25) Line 86: include primer sequences.
Now included in the method section on Line 110
26) Line 109 is a different font.
Changed using the styles
27) Line 148: define FAAS after its full name since line 151 uses the abbreviation Added in (FAAS) to Line 180
28) Line 163: “images were processed”

Line 201 changed from images were processing

29) Line 173: use μm2 for stating ROI size instead of pixel count.

Changed to um2 on Line 228

30) Line 191: Put the full method instead of citing a paper that is not even published yet, which is cited as having been previously described. Always err on the side of detailed methods and never (within reason) just cite another paper. Everybody hates that, but yet everybody has done it (present reviewer included regrettably).

We have added in some text here, as we agree, but the paper describing this mouse is currently under review at another journal and only some of the authors on the two paper are the same, so we hope that the reviewer will understand that we can’t describe the same experiments twice in two different papers. However, we have provided what we hope is enough detail for a reader to understand what was done and for them to get in touch if they would like to use this resource. See Line 75 in methods section:

All animal experiments were performed according to the UK Home Office regulations and in compliance with EU Directive 2010/63 and the UK Animals (Scientific Procedures) Act 1986. All protocols and experiments were previously approved by the Animal Welfare and Ethical Review Body (AWERB) of the University of Glasgow and were accompanied by a UK Home Office project license (PE494BE48). The Nckap1 floxed mouse strain was created using a targeting vector (PG00182_Z_4_C05) obtained from the consortium for The European Conditional Mouse Mutagenesis Program (EUCOMM) and the description of this will be published elsewhere. ES cells transfections, clone selection and injection into C57BL/6J blastocysts were performed according to standard protocols outlined in [18, 19]. Nckap1fl/fl mice were bred with Rosa26:CreERt2+ [20] and Ink4a-/- [21] to obtain the genotype Rosa26:CreERt2+; Ink4a-/-; Nckap1fl/fl. Heterozygous matings were set up for embryos. Genotyping was performed by Transnetyx (Cordova, TN, USA). Mouse embryonic fibroblasts were harvested from a pregnant female mouse at day E13.5 following the protocol outlined in [22].”

31) Line 202-203: reference format inconsistent

Changed in the text

32) Line 207: comma after MEFS

Added in the text

33) Line 226: using [28]? Put the name in front of the reference. Removed the word “using” in Line 287
34) Line 254: “ECM-coated”
Line 310
: added in the “-“

35) Line 285: “...KO cells is reminiscent...” Line 352: Changed from cells “are” reminiscent

36) Line 296-297: grammar

Sentence changed on Line 364 to “Using a specific N-WASP antibody, we analysed the endogenous localisation of N-WASP in Nckap1 KO cells that have a pseudopod-like protrusion

37) Line 312: “experiments”

Added the “s” to experiment

38) Line 340: “to have an impact on focal...” or “to impact focal...”

Changed on Line 410 to “to have an impact on focal”

39) Figure 3A: y-axis grammar

Changed in the figure

40) Figure 3C label text has a space, reads as “Immunof luorescence", Figure 3C label text there is a space in detected, reads as “det ected”

I think this is likely the quality of the figures sent as it is not in the original .ai or tiff files

41) Line 395: says violin plots present median, but there does not appear to be any indication of this on the plots.

For ease of visualisation, we changed the figure to dot plots with the grey mean and SD instead of median.

42) Figure 4A and B, there are significance bars for between-cell comparison, but the text mentions also a significant difference for the between-substrate comparison. Can this be presented as well? Given that the figure title is that collagen partly rescues the migratory defect, it seems this may be a relevant detail to show.

Stats have now been analysed and significance been added above the bars

43) Line 422: Amato et al, inconsistent reference style

Changed to numbered

44) Line 460: comma after fibroblasts (unless you want the reader to read the sentence 3 times to figure it out.)

Added a comma

45) Line 474: period after sentence missing

Added the full stop.

Round 2

Reviewer 2 Report

Whitelaw et al., have responded to almost all my comments satisfactorily. There is only one issue I have with a sentence in the abstract (and corresponding text in the main text): “ [the pseudopod] phenotype was more prevalent on collagen than fibronectin where we observed an increase in migratory speed”. The authors have not directly shown that the pseudopod phenotype is more prevalent on collagen (It can be shown by counting the number of pseudopods extended per unit time in a given number of cells). They infer it based on increased migratory speed and distance travelled. They should either modify the text or include data to support the text as it is.

Author Response

This was a very good suggestion from the reviewer and we made additions to figure 4. For this we analysed the number of Nckap1 KO cells that formed a pseudopod and migrate throughout the duration of the random migration assay. We also calculated the length of the pseudopods generated either on collagen or FN. And finally we correlated time spent in pseudopodia shape to the average speed and distance travelled. We have added sections to the methods, changes to the main text and legend.

Main text changes:

This ability of the Nckap1 KO cells to polarize and form a protrusion and thus migrate is observed on both fibronectin and collagen coated surfaces, however this migratory mode was more prevalent on collagen-coated surfaces where a higher percentage of cells formed pseudopod-like extensions (Figure 4f). Many Nckap1 KO cells form and extend more than one pseudopod. In some instances the pseudopod will split but it was observed that effective migration, one must dominate. These pseudopodia extensions formed while migrating over collagen were also significantly longer in length than those formed on fibronectin (Figure 4g). Interestingly, the pseudopod-like extensions produced by the Nckap1 KO cells had a similar range of lifetimes on both surfaces (Figure 4h, i). However, while in a pseudopod shape, Nckap1 KO cells had improved migratory capabilities over collagen, where the pseudopod-like extensions correlated with increased speed (Figure 4h) and migratory distance (Figure 4i), when compared to fibronectin.